# Expression and Regulation of CD73 during the Estrous Cycle in Mouse Uterus

**DOI:** 10.3390/ijms22179403

**Published:** 2021-08-30

**Authors:** Jihyun Lee, Haeun Park, Sohyeon Moon, Jeong-Tae Do, Kwonho Hong, Youngsok Choi

**Affiliations:** Humanized Pig Research Center, Department of Stem Cell and Regenerative Biotechnology, Konkuk University, Seoul 05029, Korea; lsw46340@naver.com (J.L.); phe325@naver.com (H.P.); 1004sh.moon@gmail.com (S.M.); dojt@konkuk.ac.kr (J.-T.D.); hongk@konkuk.ac.kr (K.H.)

**Keywords:** CD73, ecto-5′-nucleotidase, uterus, estrogen, progesterone, estrous cycle

## Abstract

Cluster of differentiation 73 (CD73, also known as ecto-5′-nucleotidase) is an enzyme that converts AMP into adenosine. CD73 is a surface enzyme bound to the outside of the plasma membrane expressed in several cells and regulates immunity and inflammation. In particular, it is known to inhibit T cell-mediated immune responses. However, the regulation of CD73 expression by hormones in the uterus is not yet clearly known. In this study, we investigated the expression of CD73 in ovariectomized mice treated with estrogen or progesterone and its regulation in the mouse uterus during the estrous cycle. The level of CD73 expression was dynamically regulated in the uterus during the estrous cycle. CD73 protein expression was high in proestrus, estrus, and diestrus, whereas it was relatively low in the metestrus stage. Immunofluorescence revealed that CD73 was predominantly expressed in the cytoplasm of the luminal and glandular epithelium and the stroma of the endometrium. The expression of CD73 in ovariectomized mice was gradually increased by progesterone treatment. However, estrogen injection did not affect its expression. Moreover, CD73 expression was increased when estrogen and progesterone were co-administered and was inhibited by the pretreatment of the progesterone receptor antagonist RU486. These findings suggest that the expression of CD73 is dynamically regulated by estrogen and progesterone in the uterine environment, and that there may be a synergistic effect of estrogen and progesterone.

## 1. Introduction

Most female mammals undergo dynamic cycle changes that occur in the reproductive system for pregnancy. They are characterized by morphological and physiological changes caused by reproductive hormones [1]. In humans, there is a reproductive cycle called the menstrual cycle, which lasts about 28 days [2]. In rodents, it is called the estrous cycle, and the estrous cycle in mice occurs over 4–5 days [3]. The estrous cycle in mice consists of four stages: proestrus, estrus, metestrus, and diestrus [4]. Each stage is changed under the influence of two major ovarian hormones, estrogen (E_2_) and progesterone (P_4_) [5]. Mouse also undergoes multiple proliferation and apoptosis processes during the estrous cycle [6].

Cluster of differentiation 73 (CD73, also known as Ecto-5′-nucleotidase) is an important enzyme that produces adenosine through ATP metabolism by working with CD39, an upstream signaling molecule. CD73 dephosphorylates extracellular AMP to adenosine [7,8]. The activity of CD73 was first demonstrated in the heart and skeletal muscle [9]. It has now been found in bacteria, plant cells, and vertebrate tissues [10].

According to a recent study, CD73 was mainly detected in stromal cells during proliferation and secretion stages, and CD73 was studied to be damaged in endometrial tissue of endometriosis [11]. In addition, studies related to CD73 and hormones have shown that positive progesterone receptor (PR) expression is correlated with CD73 expression in the breast [12,13]. These studies imply that CD73 expression is associated with steroid hormones. However, it is not yet clear whether CD73 expression is regulated by intrauterine steroid hormones. Thus, we investigated the expression of CD73 in the mouse uterus and whether CD73 is expressed and regulated by estrogen and progesterone.

## 2. Results

### 2.1. Expression of CD73 in the Mouse Uterus during Estrous Cycle

To investigate the expression of CD73 during the estrous cycle in the mouse uterus, we performed Western blot analysis. Protein was extracted from the uterus at each stage of the estrous cycle. The expression level of CD73 was dynamically regulated during the estrous cycle (Figure 1A). β-actin was used as a loading control. CD73 expression was relatively high in proestrus, estrus, and diestrus in the estrous cycle, whereas it was lowest in metestrus. The relative intensity of CD73 protein seemed to be higher than in the other stages, but there was no statistical significance (Figure 1B). These results suggest that CD73 is regulated in the uterus during the estrous cycle. Vaginal smear assay was used to identify the estrous cycle consisting of proestrus, estrus, metestrus, and diestrus. Each stage of the cycle can be determined according to the density and ratio of epithelial cells and leukocytes by checking vaginal cytology (Figure 1C).

### 2.2. The Localization of CD73 in Mouse Uterus at Each Stage of Estrous Cycle

Immunofluorescence (IF) was performed using rabbit polyclonal CD73 antibody to confirm the expression pattern of CD73 at each stage of the estrous cycle in the mouse uterus. The expression of CD73 in IF was quantified by a fluorescence microscope. IF results show that CD73 is present in all stages of the estrous cycle in the uterus (Figure 2A). It was also detected in all layers of the endometrium of the luminal epithelium (LE), glandular epithelium (GE), and stroma (S). Interestingly, CD73 is present in the cytoplasm, not the nucleus, in the luminal epithelium (LE) and glandular epithelium (GE). The relative expression of CD73 did not show a significant difference during the estrous cycle in the luminal epithelium (LE) and glandular epithelium (GE). However, in the stroma, the protein of CD73 was lowest in the metestrus stage and the highest in diestrus than in other estrous stages (Figure 2B). These results suggest that CD73 plays a role in the stroma of the endometrium during the estrous cycle.

### 2.3. Estrogen Effect on CD73 Expression in Ovariectomized (OVX) Mouse Uterus

During the estrous cycle, two major steroid hormones, estrogen and progesterone, cause uterine morphological changes. Therefore, we used ovariectomized mice to confirm the change in the expression of CD73 by estrogen treatment. Ovariectomized mice were treated with estrogen and uteri were collected at 0, 1, 6, 12, and 24 h after estrogen treatment. To confirm the expression of *CD73* mRNA, we performed RT-PCR and quantitative RT-PCR. RT-PCR analysis showed that the expression of *CD73* mRNA did not differ significantly in the time after estrogen treatment (Figure 3A). Lactoferrin (*Lf*) expression was used to confirm the estrogen response in ovariectomized mice. In semi-quantitative RT-PCR, the expression of the *CD73* transcript was lowest at 6 h after estrogen treatment, but there was no significant difference (Figure 3B). Furthermore, we investigated the level of CD73 protein after estrogen treatment using Western blot analysis (Figure 3C). The quantification of CD73 protein intensity shows that there is no difference in expression levels according to estrogen treatment after 0, 1, 6, 12, and 24 h (Figure 3D). This implies that CD73 is not significantly affected by estrogen.

### 2.4. Regulation and Localization of CD73 Expression by Progesterone

Next, since progesterone is an important hormone in the uterus like estrogen, we investigated whether the expression of CD73 is affected by progesterone. Similar to estrogen treatment, ovariectomized mice were used. Ovariectomized mice were treated with progesterone and uteri were obtained at 0, 3, 6, 12, and 24 h. Western blotting was performed to confirm the expression level of CD73. As a result, it was gradually increased after progesterone administration compared to 0 h, and CD73 expression reached the maximum level after 6 h (Figure 4A,B). Immunofluorescence (IF) showed that intrauterine CD73 was present in all layers of the endometrium (LE, GE, S) without significant differences (Figure 4C). Interestingly, the immunofluorescent localization of CD73 in hormone treated tissue after 6 h showed significant apical staining in the luminal epithelium (Figure 4C). It was also found that the CD73 protein was barely expressed without P_4_ treatment (0 h). The expression level of CD73 increased dramatically after 6 h of progesterone treatment and then decreased over time (Figure 4D). CD73 expression may be regulated by progesterone during the estrous cycle.

### 2.5. Effect of Estrogen and Progesterone on the Expression of CD73

In previous data, the expression of CD73 was not significantly affected by estrogen treatment time. However, it was shown that CD73 gradually increased after progesterone treatment and reached the maximum level after 6 h. Therefore, it was hypothesized that synergistic effects may occur when estrogen and progesterone are treated together. Therefore, the following experiment was performed to determine what effect the treatment of estrogen and progesterone would have on the expression of CD73. Since the expression of CD73 induced by progesterone treatment peaked at 6 h, the time point was set at 6 h and the uteri were collected 6 h after steroid hormone treatment. The experiment was performed using samples of the oil treatment group, progesterone-only treatment group (P_4_), estrogen and progesterone treatment group (P_4_ + E_2_), and RU486 pretreatment group before estrogen and progesterone treatment (RU/P_4_ + E_2_). The expression of CD73 in Western blot analysis by extracting protein from uteri of ovariectomized mice was increased in the progesterone-only treatment group compared to the oil treatment group, which was the same as the previous data. In addition, CD73 protein increased in the estrogen–progesterone-treated group compared to the progesterone-only treatment group and decreased again in the RU486 pretreatment group before estrogen and progesterone treatment (Figure 5A,B). When the expression pattern and localization of CD73 were confirmed by immunofluorescence results, CD73 expression was increased 6 h after progesterone treatment alone compared to the oil treatment group in the luminal epithelium (LE) and glandular epithelium (GE). This is the same as the result confirmed in the previous experiment. CD73 protein was increased in the group administered with estrogen and progesterone simultaneously compared to the group treated with progesterone alone. It was also decreased in the RU486 treatment group, a progesterone receptor antagonist, before estrogen and progesterone treatment. CD73 expression in stroma was significantly increased in the group treated with the combination of estrogen and progesterone. In the RU486 pretreatment group before estrogen and progesterone treatment, the stroma decreased more than the luminal epithelium (LE) and glandular epithelium (GE) (Figure 5C,D). These results suggest that CD73 receives synergistic effects of estrogen and progesterone.

## 3. Discussion

In this report, we investigated the expression and regulation of CD73 in the uterus. We found that the expression of CD73 changes dynamically in the mouse uterus during the estrous cycle (Figure 1 and Figure 2). Interestingly, CD73 expression was increased when progesterone or both progesterone and estrogen were administered simultaneously rather than estrogen alone, and its immunostaining images showed the apical location of CD73 in the luminal epithelium of OVX after hormone treatment (Figure 4 and Figure 5). This indicates that CD73 plays a role in the uterine dynamics during the estrous cycle. Furthermore, this suggests that there is a synergistic effect of estrogen and progesterone on the expression of CD73.

CD73 is a protein present as a dimer bound to the plasma membrane surface by glycosyl-phosphatidyl inositol [14,15]. CD73 is a metabolic enzyme that hydrolyzes AMP into extracellular adenosine and is known to regulate immunity and inflammation [16]. Adenosine produced by CD73 activity can promote tumor cell proliferation by binding to A2a and A2b receptors expressed in tumor cells. Conversely, by activating immunosuppressive cells such as T-regulatory cells (Treg) and tumor-related macrophages, anti-tumor activity can be improved [17]. It is also known that CD73 is expressed in CD4 effector cells (anergic CD4 + T cells) to maintain autoimmunity in heathy tissue and to protect the fetus from maternal immunity during pregnancy [18,19]. Recent study showed that NK cells transport CD73 to the cell undergoing transcriptional reprogramming to define regulatory NK cells in the tumor microenvironment [20]. In fact, the regulation of the intrauterine microenvironment is important for preparing pregnancy by immunosuppression. It can be inferred that these results are related to immunosuppression in the uterus.

CD73 expression is ubiquitous. It is expressed in most tissues such as brain [21,22], kidney [23,24], liver [25,26], lung [27,28], and heart [29,30]. This supports the idea that CD73 plays a role in cell type or tissue-dependency. Many reports provide a glimpse into the role and molecular mechanism of CD73 in the uterine endometrium. In the kidney, CD73 induction protects against renal ischemia and reperfusion injury [31]. In the lung, CD73 prevents alveolar damage in hyperoxia [28]. The upregulation of CD73 showed cardioprotective effects in heart failure [32]. Additionally, CD73 is involved in regulating hepatic fibrosis [26].

Recent data explained that the regulatory mechanism of CD73 for liver fibrosis is through inhibiting the activation and proliferation of stellate cells [33]. They reported that the Wnt/β-catenin signaling pathway is related to this mechanism [33]. In fact, numerous Wnt signaling members, Wnt2, Wnt3, Wnt4, Wnt5, Wnt7, Wnt9, Wnt10, Wnt11, and Wnt16, are known to be critical for both embryonic development and endometrium dynamics before implantation [34]. This gives us a clue for understanding its mechanism in the uterus.

The uterus is the site where the implantation of the fertilized egg, placenta development, and embryonic development occur. ATP, which is released by autocrine or paracrine release from epithelial cells of the uterus, plays an important role in regulating endometrial functions such as sperm migration and the implantation of fertilized eggs [35]. Extracellular ATP and adenosine act through purine receptors, affecting the endometrial fluid microenvironment and regulating female fertility. In addition, extracellular adenosine plays a signaling role in controlling early morphogenesis after implantation [36] and regulates uterine muscle contraction [37]. CD73 (ecto-5′-nucleotidase) is mainly present in the endometrial epithelia, both luminal and glandular, and their expression changes even after menopause [38].

According to recent studies, it is known that the expression of CD73 is impaired in ectopic and eutopic endometrial stroma [11]. In addition, the activity of CD73 was much higher in endometrial tumors than in simple cysts and could be quantified in ovarian cyst aspiration [39]. This indicates that CD73 may be involved in endometrial disease. Therefore, we investigated the relationship between the expression of CD73 in the uterus and hormones. The estrous cycle is divided into four stages: proestrus, estrus, metestrus, and diestrus, and changes occur repeatedly by steroid hormones such as estrogen and progesterone. Diestrus represents the progesterone-dominant phase during the estrous cycle, proestrus represents the transition period from the progesterone-dominant diestrus phase to estrogen dominance, whereas metestrus represents the opposite change in the endocrine environment [40]. CD73 expression is also highest in the diestrus stage, which is the progesterone-dominant stage among the estrous cycle stages. This suggests that CD73 expression is correlated with progesterone. In the metestrus stage, low levels of estrogen and progesterone and low CD73 expression appear to be related. This suggests that CD73 is associated with two steroid hormones, especially when both hormones are involved simultaneously. To determine whether these hormonal changes act alone, the expression of CD73 in the uterus was shown using the OVX model.

The endometrium is altered by steroid hormones, and CD73 is particularly affected by progesterone. It was increased when progesterone was administered to OVX mice. Interestingly, the expression of CD73 was increased in the estrogen–progesterone-treated group compared to the progesterone-only group. This suggests that there may be a synergistic effect of estrogen and progesterone. In addition, CD73 expression was reduced by blocking progesterone receptors. This means that the expression of CD73 is regulated in the uterus by progesterone and progesterone receptor signals. In conclusion, this study showed that CD73 can be dynamically regulated by estrogen and progesterone in the uterus, and that there may be a synergistic effect of estrogen and progesterone.

## 4. Materials and Methods

### 4.1. Animal Care and Experimentation

Animal experiments were performed using 6- to 7-week-old ICR mice provided by JA BIO (Suwon, Korea). Mice were housed under controlled temperature and lighting conditions. A light was lit for 12 h every day and mouse was fed ad libitum. Animal care and use were carried out according to the guidelines for the care and use of laboratory animals and were approved by the Institutional Animal Care and Use Committee (IACUC) of Konkuk University (approval number, KU19216).

### 4.2. Confirmation of Estrous Cycle and Uterus Sampling

The mouse estrous cycle was determined through the vaginal smear assay as in previous studies [41,42]. Vagina was gently flushed with DPBS and sucked in, which was repeated 5 times. We put the collected DPBS on a glass slide as a drop, dried it on a heat block, and dyed it with hematoxylin (Vector Laboratory, Burlingame, CA, USA) for 1 min. We rinsed it with tap water for 5 min and incubated it in 50%, 75%, and 90% ethanol for 5 min. After treatment with eosin Y (Sigma-Aldrich, St. Louis, MO, USA), the stained slides were immersed in 90% ethanol and 100% ethanol for 5 min. After incubation in Xylene, slides were mounted with a Permount mounting medium (Thermo Fisher Scientific, Waltham, MA, USA). Staining was observed using a microscope, and each stage of the estrous cycle was identified by vaginal smear cytology. After determining the estrous cycle, the uteri in the proestrus, estrus, metestrus, and diestrus stages were collected. One of two uterine horns was used for RNA and protein extraction, and the other was fixed in 4% paraformaldehyde (PFA) for paraffin blocks.

### 4.3. Ovariectomy and Hormone Injection

To investigate the effects of estrogen and progesterone on CD73 expression in the uterus, 7~8-week-old ICR mice were used to perform ovariectomy as in previous studies [43,44]. Seven~eight-week-old ICR mice were anesthetized by intraperitoneal injection of 2,2,2-Tribromoethanol (Avertin) (Sigma-Aldrich, St. Louis, MO, USA) or Alfaxalone (Careside, Gyeonggi-do, Korea). We cut the skin slightly with sterile surgical tools to find the ovaries under the fat and cut them. Then, we used a sterile suture (Ailee Co., Busan, Korea) to suture. We disinfected with fovidin (Firson Co Ltd., Cheonan-si, Korea) and placed it on a 42 °C heat warmer to watch the condition until awakening from anesthesia. After waking up from anesthesia, water was given, followed by a stabilization period. After a 2-week stabilization period, ovariectomized mice were injected subcutaneously with β-estradiol (E_2_, 200 ng/mouse, Sigma-Aldrich, St. Louis, MO, USA) or progesterone (P_4_, 2 mg/mouse, Sigma-Aldrich, St. Louis, MO, USA). The mice were sacrificed and their uteri were collected at various time points (0, 1, 3, 6, 12, or 24 h) after hormone treatment. Additionally, ovariectomized mice were injected with progesterone priming prior to a second injection of P_4_, and E_2_ as described previously [43,44]. To investigate whether the expression of CD73 in the uterus is dependent on progesterone, a progesterone receptor antagonist, RU486 (1 mg/mouse, Sigma-Aldrich, St. Louis, MO, USA), was pretreated 30 min prior to progesterone and estrogen injection. Sesame oil (100 μL/mouse, Sigma-Aldrich, St. Louis, MO, USA) was used for control mice.

### 4.4. RNA Preparation, Reverse Transcription PCR (RT-PCR), and Quantitative Real-Time PCR (qRT-PCR)

To isolate total RNAs, tissue and cells were homogenized using a Trizol reagent (Invitrogen, Waltham, MA, USA) and a homogenizer and purified with chloroform (Sigma-Aldrich, St. Louis, MO, USA). Total RNAs (1 μg) were reverse-transcribed using the SensiFast^TM^ cDNA synthesis kit (Bioline, London, UK) according the manufacturer’s protocol. The product cDNAs were used in RT-PCR and quantitative RT-PCR analyses. RT-PCRs were performed using the ProFlex PCR system (Applied Biosystems, Foster City, CA, USA). The PCR conditions and the size of the gene-specific primers are shown in Table 1. Ribosomal protein L7 (*Rpl7*) was used as a reference gene, and lactoferrin (*Lf*) was used for an indicator of hormone replacement after ovariectomy. PCR products were analyzed by gel electrophoresis on a 2% agarose gel using a ChemiDoc^TM^ XRS + system (Bio-Rad Life Sciences, Hercules, CA, USA). Quantitative real-time PCR (qRT-PCR) analysis was performed using iQ^TM^ SYBR^®^ Green Supermix (Bio-Rad Life Sciences, Hercules, CA, USA) and QuantStudio 1 Real-Time PCR Instrument (Applied Biosystems, Foster City, CA, USA). Gene expression levels were calculated by comparative C_T_ (ΔΔC_T_) method. Relative gene expression was normalized to the C_T_ value of *Rpl7* to obtain ΔΔC_T_ value.

### 4.5. Protein Preparation and Western Blot Analysis

After freezing the uterine sample in liquid nitrogen, it was homogenized using a homogenizer. Protein was extracted using RIPA lysis buffer (Sigma-Aldrich, St. Louis, MO, USA) containing protease inhibitor cocktail (Roche Applied Sciences, Indianapolis, IN, USA). It was incubated for 30 min while vortexing every 10 min on ice. After incubation, the sample was centrifuged at 4 °C, 13,000 rpm for 30 min. The supernatant with protein was collected and used. The whole cell extracts were prepared and analyzed by Western blotting. Their concentration was measured by Bradford assay (Thermo Fisher Scientific, Waltham, MA, USA). A total of 20 μg of total protein was loaded for each lane and separated by loading on SDS-PAGE (10% gradient gel), and then transferred to PVDF (polyvinylidene difluoride) membranes (Bio-Rad Life Sciences, Hercules, CA, USA). Membranes were blocked with 5% Difco^TM^ Skim milk (BD Biosciences, Franklin, NJ, USA) in PBS containing 0.05% Tween-20 (PBS-T) for 1 h at room temperature. The membrane was incubated overnight at 4 °C with anti-CD73 rabbit monoclonal antibody (1:1000 dilution, D7F9A, Cell signaling technology, Danvers, MA, USA) diluted in 5% skim milk. It was washed three times with PBS-T and treated with HRP-conjugated secondary antibody (1:10,000 dilution, TA140003, Origene, Rockville, MD, USA) at RT for 1 h. The membrane was developed using Western Femto ECL kit (LPS solution, Daejeon-si, Korea). Relative protein expression was analyzed by ChemiDoc^TM^ XRS + system (Bio-Rad Life Sciences, Hercules, CA, USA). To confirm whether the protein was quantified in the same amount for each sample, β-actin antibody (1:10,000 dilution, sc-47778, Santa Cruz Biotechnology Inc., Dallas, TX, USA) was used as a loading control on the same blot, which was treated with 250 μL of 10% sodium azide in 20 mL of buffer solution before reuse. One-way ANOVA analysis was performed by Excel software.

### 4.6. Immunofluorescence

Uteri collected from mice were fixed in 4% paraformaldehyde (PFA) at 4 °C for at least 1 week. After embedding uteri into paraffin, the uterine paraffin blocks were sectioned using a microtome to a thickness of 5μm. Deparaffinization was performed in xylene twice for 10 min each, rehydration was performed in 100%, 90%, 75% ethanol and 50% ethanol for 5 min, and then they were washed with tap water for 5 min. Using an antigen retrieval steamer (IHC world, Gyeonggi-do, Korea), they were boiled in the antigen retrieval buffer (10 mM sodium citrate, 0.05% Tween20, PH 6.0) and cooled at room temperature for 30 min. After washing three times with tap water and PBS-T (PBS containing 0.05% tween20) for 5 min, a hydrophobic wall was drawn around the tissue with an ImmEdge pen (Vector Labs, Burlingame, CA, USA). The slides were blocked with PBS-T containing 4% BSA and 5% normal goat serum at room temperature for 3 h. Then, sections were treated with anti-CD73 rabbit polyclonal antibody (1:1000 dilution, ab175396, Abcam, Cambridge, UK) overnight at 4 °C. After washing three times with PBS-T, secondary antibody conjugated with Alexa-Fluor antibody 546 (Thermo Fisher Scientific, Waltham, MA, USA) was incubated for 1 h at RT. The slide was washed with PBS-T and treated with DAPI (4’,6-Diamidino-2-phenylindole) (1:10,000 dilution, D3571, Thermo Fisher Scientific, Waltham, MA, USA) for 10 min, and then mounted on cover slip with mounting medium (DAKO, Glostrup, Denmark). Slides were observed using a confocal microscope. Additionally, we showed background staining using only the secondary antibody as negative control on the uterine section of the stage showing the most staining. To quantify the relative intensity, four images were randomly selected and analyzed using ZEN 2012 (Carl Zeiss Co. Ltd., Oberkochen, Germany) imaging software and ImageJ.

### 4.7. Statistical Analysis

All experimental data are reported as mean ± SEM (standard error of mean). Results were analyzed using a one-way ANOVA for statistical evaluation. A *p*-value less than 0.05 was considered statistically significant.

## Figures and Tables

**Figure 1 ijms-22-09403-f001:**
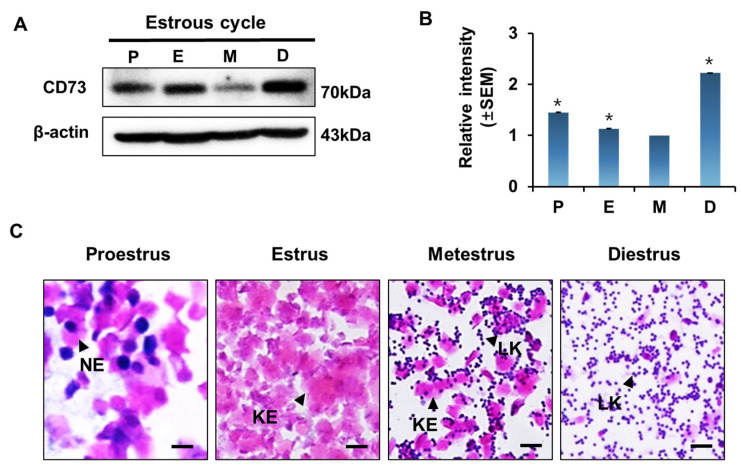
CD73 expression in the mouse uterus during estrous cycle. (**A**) Western blot analysis of CD73 in mouse uterus at each stage during estrous cycle (*n* = 4). β-actin antibody was used for loading control. P, proestrus; E, estrus; M, metestrus; D, diestrus. (**B**) Relative levels of CD73 in uterus from different stages of estrous cycle. Each value is expressed as a relative value based on metestrus stage of estrous cycle. Data are shown with mean ± SEM. The one-way ANOVA analysis was used to calculate *p*-value, * *p*-value < 0.01. (**C**) Vaginal smear assay was performed to confirm the estrous cycle. NE, nucleated epithelial cell; KE, anucleated keratinized epithelial cell; LK, leukocyte. The black scale bars indicate 50 μm.

**Figure 2 ijms-22-09403-f002:**
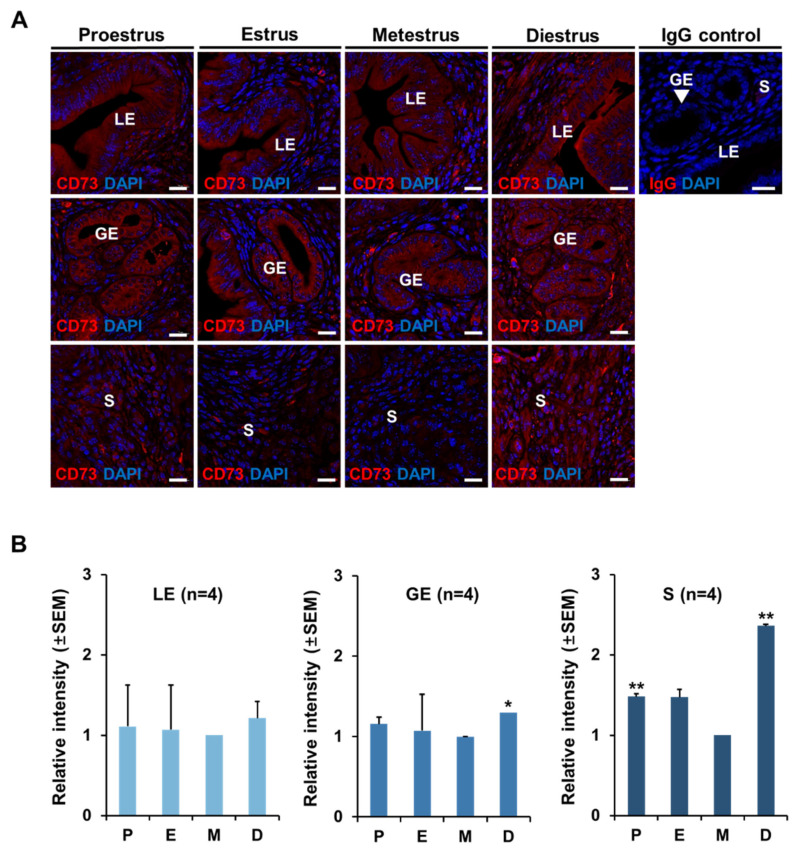
CD73 localization in mouse uterus at each stage of estrous cycle. (**A**) Immunofluorescence analysis of CD73 at four stages of estrous cycle (each group *n* = 4). Blue color represents DAPI in the uterus. Normal rabbit IgG (IgG control) was used as a negative control. LE, luminal epithelium; GE, glandular epithelium; S, stroma. The white scale bars indicate 20 μm. (**B**) Quantitation of the relative levels of CD73 in mouse uterus during estrous cycle using imaging software Carl Zeiss ZEN 2012 in luminal epithelium (LE), glandular epithelium (GE), and stroma (S), respectively. Each value is expressed as a relative value based on metestrus stage of estrous cycle. Four animals in each group were examined. P, proestrus; E, estrus; M, metestrus; D, diestrus. The one-way ANOVA analysis was used to calculate *p*-value, * *p*-value < 0.01, ** *p*-value < 0.05.

**Figure 3 ijms-22-09403-f003:**
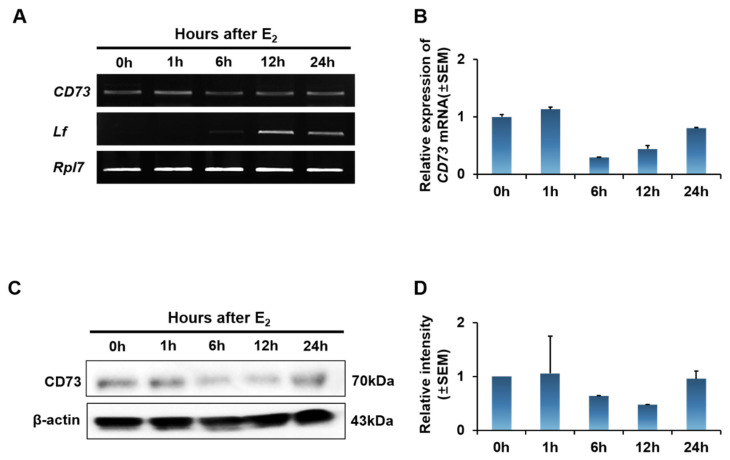
Estrogen effect in CD73 expression in ovariectomized (OVX) model. (**A**) RT-PCR analysis of *CD73* mRNA in the uterus of the OVX mice after E_2_ treatment for 0, 1, 6, 12, and 24 h (each group *n* = 4). Lactoferrin (*Lf*) gene was used to confirm the response of estrogen after ovariectomy. Ribosomal protein L7 (*Rpl7*) gene was used as an internal control. (**B**) Quantitative RT-PCR shows the relative fold change of *CD73* expression after estrogen treatment. The relative expression value was based on the value at 0 h after E_2_ treatment. (**C**) Expression of CD73 was analyzed by Western blot analysis in the uterus of the OVX mice after E_2_ treatment for 0, 1, 6, 12, and 24 h (each group *n* = 4). β-actin was used as an internal control. (**D**) Relative level of CD73 protein intensity in uterus of OVX mice after E_2_ treatment. The relative expression value was based on the value at 0 h after E_2_ treatment. Data are presented as the mean intensity ± SEM.

**Figure 4 ijms-22-09403-f004:**
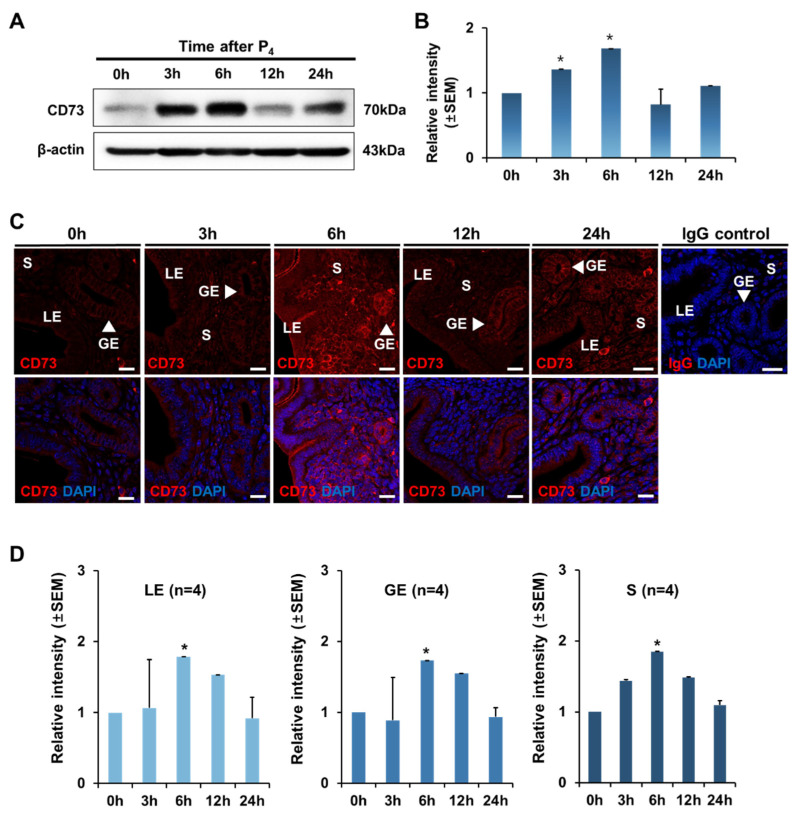
The effect of progesterone on CD73 expression in ovariectomized (OVX) mouse uterus. (**A**) Western blot analysis for relative levels of CD73 in the uterus of the OVX mice after P_4_ treatment for 0, 3, 6, 12, and 24 h (each group *n* = 4). β-actin was used as a loading control. (**B**) Quantitative Western blot analysis showed the relative expression of CD73 in the uterus of the OVX mice after P_4_ treatment. The one-way ANOVA analysis was used to calculate *p*-value, * *p*-value < 0.01. (**C**) Confocal microscopic images represent the localization and expression level of CD73 (red) and DAPI (blue) in the uterus of the OVX mice treated with P_4_. Normal rabbit IgG (IgG control) was used as a negative control for a secondary antibody. LE, luminal epithelium; GE, glandular epithelium; S, stroma. The white scale bars indicate 20 μm. (**D**) Quantitation of the relative levels of CD73 in the uterus of P_4_-treated OVX mice using imaging software Carl Zeiss ZEN 2012 in luminal epithelium (LE), glandular epithelium (GE), and stroma (S), respectively. The relative expression value was based on the value at 0 h after P_4_ treatment. Four animals in each group were examined. The one-way ANOVA analysis was used to calculate *p*-value, * *p*-value < 0.01.

**Figure 5 ijms-22-09403-f005:**
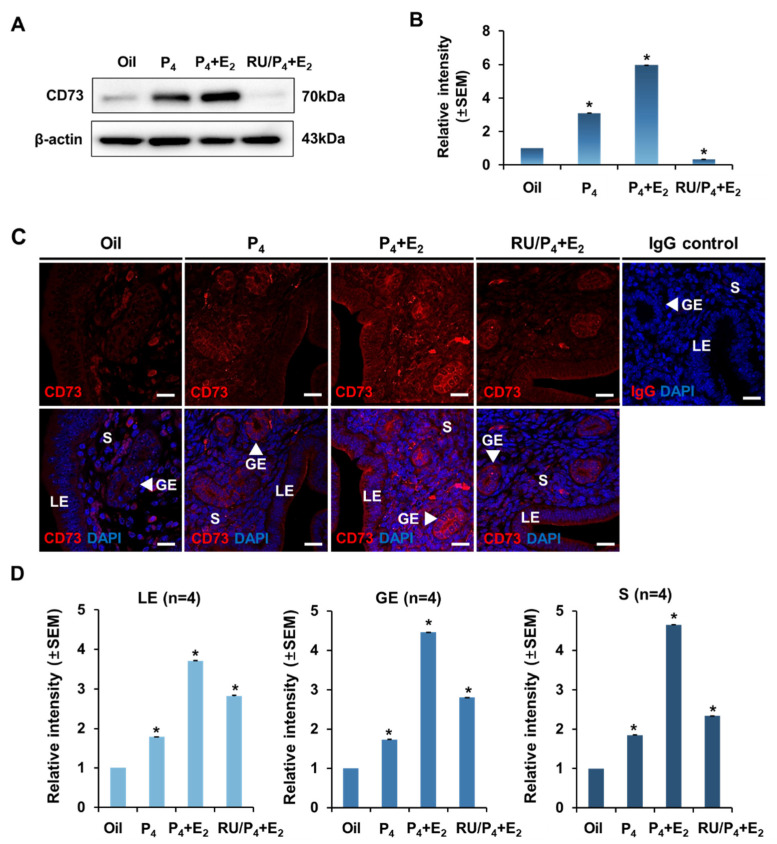
Expression of CD73 during simultaneous treatment of estrogen and progesterone. (**A**) Western blot analysis showed the expression of CD73 in the uterus of the OVX mouse treated with oil, P_4_ + E_2_, or RU486 + P_4_ + E_2_ (RU/P_4_ + E_2_) (each group *n* = 4). β-actin was used for an internal control. (**B**) Relative protein expression of CD73 in the uterus of the OVX mice treated with oil, P_4_ + E_2_, or RU486 + P_4_ + E_2_ (RU/P_4_ + E_2_). The relative expression value was based on the oil treatment value. Data were normalized to β-actin expression and presented as the mean intensity ± SEM. * *p*-value < 0.01. (**C**) Confocal microscopic images represent the localization and expression level of CD73 (red) and DAPI (blue) in the uterus of the OVX mice treated with oil, P_4_ + E_2_, or RU486+ P_4_ + E_2_ (RU/P_4_ + E_2_). Normal rabbit IgG (IgG control) was used as a negative control for a secondary antibody. LE, luminal epithelium; GE, glandular epithelium; S, stroma. The white scale bars indicate 20 μm. (**D**) Quantitative immunofluorescence analysis showed the relative expression of CD73 in the uterus of OVX mice treated with oil, P_4_ + E_2_, or RU486+ P_4_ + E_2_ (RU/P_4_ + E_2_). Relative intensity was analyzed by imaging software Carl Zeiss ZEN 2012 in luminal epithelium (LE), glandular epithelium (GE), and stroma (S), respectively. The relative expression value was based on the oil treatment value. Four animals in each group were examined. Data are shown with mean ± SEM. * *p*-value < 0.01.

**Table 1 ijms-22-09403-t001:** Primer sequences and RT-PCR conditions.

Genes	Accession No.	Primer Sequence	Product Size (bp)
*CD73*	NM_011851.4	^1^ F; AGGTTGTGGGGATTGTTGGA ^2^ R; CCCCAGGGCGATGATCTTAT	152
*Rpl7*	NM_011291.5	^1^ F; TCAATGGAGTAAGCCCAAAG ^2^ R; GAAGAGACCGAGCAATCAAG	246
*Lf*	NM_008522.3	^1^ F; AGGAAAGCCCCCCTACAAAC ^2^ R; GGAACACAGCTCTTTGAGAA	141

^1^ F; forward primer, ^2^ R; reverse primer.

## Data Availability

Not applicable.

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
