# Peer review of "Expression and Regulation of CD73 during the Estrous Cycle in Mouse Uterus"

_ijms, 2021, doi:10.3390/ijms22179403_

Round 1
Reviewer 1 Report
Dear Editor and Authors, I thank you for the opportunity to read and give my opinion on this study. This article studied the expression of CD73 in mouse uterus and found that it was dynamically regulated mainly by progesterone in the uterine environment, and that there may be a synergistic effect of estrogen and progesterone.
I have several questions/comments.
- I found that some results in a reference are inconsistent with yours, titled “Changes in expression and activity levels of ecto-5’-nucleotidase/CD73 along the mouse female estrous cycle” (https://doi.org/10.1111/j.1748-1716.2010.02095.x). In this article, the immunohistochemistry showed that maximum expression of CD73 in uterus was detected at the estrus phase. The stroma in estrus displayed faint immunoreactivity and become maximum in metestrus. No expression was detected in glands at metestrus and diestrus. Please explain this phenomenon.
- I was curious about whether the level of estrogen and progesterone are changing with CD73 during the estrous cycle. Could you please test the level of estrogen and progesterone at each period of estrous cycle? This may be better to demonstrate the relationship between hormones and CD73.
- Have you ever tried to explore the role of molecules in the estrogen and progesterone pathway played in the regulation of CD73? This study lacks the research on molecular mechanism.
Author Response
Please find an attached file.

Reviewer 2 Report
This manuscript describes the localisation and quantification of CD73 in the mouse uterus during the estrous cycle as well as in ovariectomised (ovex) animals with exogenous hormone administration. There are parts of this manuscript which require editing and language correction and some major questions about the scientific method and results interpretation.
My issues with the manuscript are as follows
The last sentence in the abstract is too far reaching – there is no relation shown between hormone changes like induced through the ovex with hormone administration and endometriosis. This sentence should be removed.
The introduction could benefit from some language improvement and editing – eg ‘to’ should be ‘two’. There are also several instances of ‘it shows…’ please define what ‘it’ refers to
Results – there are several issues with the results section
The term ‘most expressed’ is troubling, what does this mean?
Careful using the term ‘expression’ when referring to immunofluorescent localisation – expression should be confined to RNA analysis, not protein localisation
The image of the vaginal smearing for proestrus is not correct – there should be no leukocytes in a smear at this time, just small nucleated cells with some larger keratinised cells. If these animals were processed as proestrus then this is incorrect and would invalidate the results
For all of the results figure legends it should be made clear what the significant values are compared to.
The time after hormone administration for collection of tissue should be made clear in the methods section
How was the immunofluorescent staining intensity measured and graphed? Which computer program was used, how was thresholding established? Why do these graphs measure ‘relative intensity’?
The immunofluorescent localisation of CD73 in hormone treated tissue after 6 hrs show significant apical staining in the luminal epithelium – this should be documented and discussed in the results and discussion sections
The figure legend for fig 4 discusses red staining – there is no such staining in the immunofluorescence images. Was there any manipulation of the images if so this should be stated in the methods section. There is no ** significance in this graph however this is stated in the figure legend.
Discussion is relatively well written however the last sentence about a possible biomarker for endometriosis is too far reaching for the results described in this study. Further information about luminal epithelial staining 6hrs post could be included here
Material and methods – this was the most concerning part of the this manuscript and requires extensive language correction. Some other issue about this section are as follows
The duration of the hormone injections and how the hormones are combined should be explained. It is usual for more than 1 day of hormones to be administered before sacrifice of the animals. The combination of E and P is usually P+P+PE, not just a single dose of P+E. Hormone concentrations should be in mg/kg, not per animal. A reference needs to be included to justify the hormone concentration used
The other genes of interest need to be explained in the materials and methods section
Western blot – how much protein was loaded onto the gel, how was the protein conc determined
How was the b-actin measured, was the blot stripped and reprobed?
Controls need to be explained in the immunofluorescence section
It is described that an alexa-fluor 546 conjugated secondary antibody was used, however the images show it used a different fluorophore (AF546 is orange!) If image manipulation was used this needs to be described.
Author Response
First of all, thanks for your points and comments. We know that our results are not sufficient for the conclusion. Nevertheless, we tried our best to answer. We hope that our responses have been a little more reliable. Please find an attached file for the response.

Reviewer 3 Report
In this manuscript the presented data are sound. However, there are a few points that need to be addressed and modify in the manuscript before publishing: Major revision.
- Abstract: The conclusion line ‘This suggests…….proliferated by hormonal changes’ should be rewritten.
- Keywords are not in alphabetical order.
- Line No. 36 page No. 2 “under the influence of ‘to’. The ‘to’ here should be changed to “two” or number “2”.
- The introduction was irrelevantly drafted. The role of CD73 in cancer cell proliferation and apoptosis was described but not in relation to the estrus cycle. Authors should rewrite the introduction.
- Representation of CD73 should be uniform all over the study.
- What do you interpret with the lactoferrin expression in your study?
- Why oil treatment group was included as control?
- Discussion part should be rewritten as it doesn’t discuss other authors’ findings in relation to current work.
- Should avoid repeated sentences in introduction, results, and discussion sections.
- Need to provide University animal care ethical committee approval number.
- Page No. 10, line No. 281-282 should be rewritten.
- In page No. 10, line No. 288, replace the word “leather” with “skin”.
- Authors should explain the role of Lf gene in qPCR method.
- What is the significance of the study?
- Authors should check the conclusion part: Are the results really supporting the conclusion? The tested /selected parameters are sufficient for the conclusion?
- References should be cited by following journal style/format.
- Need to check for typographical errors, plagiarism, punctuation, and grammar throughout the manuscript.
Author Response

(The authors gave the same response as above.)

Round 2
Reviewer 1 Report
The reviewer was satisfied with the revision and suggested accepting the manuscript in its current version. Congratulations to the authors for their nice work.
Reviewer 2 Report
I am happy with the corrections made by the authors. Some editing is still required in the newly incorporated sections of the manuscript.